# Differentiation of River Sediments Fractions in UAV Aerial Images by Convolution Neural Network

**Hitoshi Takechi** [1], **Shunsuke Aragaki** [2] and **Mitsuteru Irie** [3,*]

1   Blue Innovation Co., Ltd., Tokyo 113-0033, Japan; hitoshi.takechi@blue-i.co.jp
2   Department of Civil and Environmental Engineering, School of Engineering, University of Miyazaki, Miyazaki 889-2192, Japan; hh18001@student.miyazaki-u.ac.jp
3   Faculty of Engineering, University of Miyazaki, Miyazaki 889-2192, Japan
*   Correspondence: irie.mitsuteru.p2@cc.miyazaki-u.ac.jp; Tel.: +81-2958-7341

**Abstract:** Riverbed material has multiple functions in river ecosystems, such as habitats, feeding grounds, spawning grounds, and shelters for aquatic organisms, and particle size of riverbed material reflects the tractive force of the channel flow. Therefore, regular surveys of riverbed material are conducted for environmental protection and river flood control projects. The field method is the most conventional riverbed material survey. However, conventional surveys of particle size of riverbed material require much labor, time, and cost to collect material on site. Furthermore, its spatial representativeness is also a problem because of the limited survey area against a wide riverbank. As a further solution to these problems, in this study, we tried an automatic classification of riverbed conditions using aerial photography with an unmanned aerial vehicle (UAV) and image recognition with artificial intelligence (AI) to improve survey efficiency. Due to using AI for image processing, a large number of images can be handled regardless of whether they are of fine or coarse particles. We tried a classification of aerial riverbed images that have the difference of particle size characteristics with a convolutional neural network (CNN). GoogLeNet, Alexnet, VGG-16 and ResNet, the common pre-trained networks, were retrained to perform the new task with the 70 riverbed images using transfer learning. Among the networks tested, GoogleNet showed the best performance for this study. The overall accuracy of the image classification reached 95.4%. On the other hand, it was supposed that shadows of the gravels caused the error of the classification. The network retrained with the images taken in the uniform temporal period gives higher accuracy for classifying the images taken in the same period as the training data. The results suggest the potential of evaluating riverbed materials using aerial photography with UAV and image recognition with CNN.

**Keywords:** channel bed condition; particle size; convolution neural network; UAV

## 1. Introduction

The sand, gravel, and rock that make up riverbed material have multiple functions as habitats, feeding grounds, spawning grounds, and shelters for aquatic creatures [1,2]. Besides, the particle size of the riverbed material reflects the tractive force of the channel flow. By observing the riverbed, we can better understand the spatial distribution of flow velocity [3,4]. Thus, the spatial distribution of particle size of riverbed material is very important information for flood control and environmental protection in river management. Therefore, accurate information on the topography of the riverbed and its material as well as rainfall and discharge, is necessary.

The most general conventional sampling of surveying riverbed material is by field methods, such as grid-by-number or volumetric methods [5,6]. Field methods require huge labor, time, and cost because samples are collected on-site and measured or sieved. Therefore, at the site of sediment management, it is common to evaluate the riverbed condition of sandbars of several thousand square meters by conducting a quadrat survey

(using 0.5 m × 0.5 m quadrats) of a few to several tens of points. In such cases, spatial representativeness is a serious problem.

To solve those problems, a riverbed material survey using image analysis has attracted attention [7,8]. Observations of the particles on the ground surface were carried out by geomorphologists and sedimentologists, with recent work utilizing technological advances to determine grain size characteristics, using automated grain size measurements from images and geostatistical techniques combined with empirical calibration, mainly of exposed gravel bed rivers [9–11]. These methods needs to discriminate a particle to measure thus that the sand and small gravel that cannot be identified as a particle on a digital image could not be counted.

Image recognition by machine learning and deep learning has made rapid progress over a short period. Some of the applications where deep learning is used in computer vision include industrial [12–14] and medicinal [15–18] fields. A number of the research using deep learning for estimating particle size characteristics are automatically finding target objects in the image by the recognition, measuring the size of the objects, and calculating particle size distribution [13,14]. These approaches provide precise particle distribution while it takes a longer calculation time. However, the particle size information required for river management is often only representative particle size but a huge number of images covering a wide study site have to be analyzed.

Even among the researchers handling remote sensing, especially for the land use/land cover (LULC) classification, the application of AI technologies has a good affinity for the classification based on the multispectral data such as Landsat and Sentinel [19,20]. However, these studies have a larger scale of the study site and resolution than our study. Suzuki et al. [21] tried a vegetation classification using UAV and machine learning, in which they combined automatic generation of a wide-area mosaic image and vegetation classification using superpixel segmentation and machine learning. This study showed that it was possible to classify the UAV aerial image by machine learning to draw the thematic map of the defined classes. This study is similar to the spatial scale and resolution of our case.

On the other hand, Takahashi et al. [22] proposed a novel AI disease-staging system for grading diabetic retinopathy and another AI that directly suggests treatments and determines prognoses. The system is based on the convolutional neural network (CNN) with transfer learning simply and rapidly grading the images of the patients by detecting features. There was not any process measuring the length or area of objects in the image in the method.

The purpose of the present paper is to develop an evaluation method for the spatial distribution of riverbed materials in a wide area that requires a lot of classification processes of images with a shorter time. CNN can be a powerful tool for this task. We aim to classify the images into three categories: fine gravel and coarse sand, small gravel, and medium gravel (Table 1). These are defined based on the classification criteria set by the Public Works Research Institute, Japan, from the viewpoint of river development [23]. The methodology proposed in this study differs from the measuring processes of particle size, mentioned above. The riverbed images are classified based on the features detected by CNN without discrimination of independent particles and measuring them.

**Table 1.** Classification criteria in this study.

| Classification | Particle Size (mm) | Classification Name in This Study |
| --- | --- | --- |
| Medium gravel | 24.5–64 | Class 1 |
| Small gravel | 2–24.5 | Class 2 |
| Fine gravel and coarse sand | 0.5–2 | Class 3 |

## 2. Related Studies

Yamazaki et al. [24] tried image analysis instead of sampling and measuring or sieving. They manually took photographs from a tripod and measured particle size, grain-by-grain,

on the images. The shooting angles of the images were not vertical thus that geometric correction was required for each image. This image editing and counting of the grains required a large amount of labor. They confirmed that the particle size distributions obtained by image analysis were similar to those obtained by sieving. However, the sand of 20 mm or less was not recognized because of low resolution. Okada et al. [25] applied image analysis underwater and photographed a riverbed from a boat by suspending a camera from a rope. Representative particle sizes were calculated by the image analysis software. Representative particle sizes obtained by this method did not significantly differ from those obtained by the conventional method. Besides, this survey method was better for safety, economy, and the river environment because of non-contact. BASEGRAIN is image analysis software for gravel grains and was developed by Detert and Weitbrecht [26]. It can identify individual gravel particles on images, measure major and minor axes, and calculate particle size distribution. It has the advantage of providing a detailed and accurate particle-size accumulation curve that is close to that obtained by field methods because the software measures all grains that can be discriminated in an image. Harada et al. [27] compared particle size distribution for BASEGRAIN and the grid-by-number method and found that the two particle size distributions almost coincided for particle sizes larger than discriminable particle size. All of these studies measuring the size of the distinguished particle gave precise particle size distribution but need careful attention to the cause of the error, such as shooting angle, the distance between camera and ground surface, etc.

Terada et al. [28] used an unmanned aerial vehicle (UAV) and BASEGRAIN to spatially measure particle size distribution in river sandbars and compared particle size distribution before and after flooding. The size of the study site was 210 m × 410 m, the flight altitude of the UAV was 30 m, and about 200 images were photographed. Then, these images were combined to create a single overall image. This overall image was divided into 22 × 43 images with cartesian grids, and the images extracted from each grid were analyzed using BASEGRAIN, and the particle size of each stone was measured precisely. As a result, it was confirmed that it was possible to obtain the detailed spatial particle size distribution over a wide area and movement of surface sediment before and after flooding. However, there is much labor to get such spatial distribution information even with the moderate resolution over a wide area because image software such as BASEGRAIN analyzes images one-by-one. Besides, there is a disadvantage that fine particle components that cannot be recognized on images are not discriminated, and must be extrapolated from a large particle size distribution. On the other hand, the classification criterion required for the actual sediment management is grouping into a smaller number of categories, such as rock, gravel, and sand, as this study adopted.

## 3. Materials and Methods

The method that we propose can be structured according to the following steps: aerial photography using UAV, sieving, image preprocessing (clipping the area for the image analysis), BASEGRAIN analysis, and image classification by CNN.

### 3.1. Study Area

Mimikawa River located in Miyazaki Prefecture, Japan, has a length of 94.8 km and a watershed area of 884.1 km$^2$. It has 7 dams, developed between the 1920s and the 1960s, whose purpose is only power generation, not flood control (Figure 1). However, sedimentation in the reservoirs breaks the natural transportation of solid matter along the channel. Just downstream of the reservoir, especially, the mother rock emerges, and gravel and sand are removed from the river bed by intermittent flood discharge. Degradation of the flow regime with a large gap between maximum and usual flow rates and low diversity of bed conditions has significantly affected the ecosystem. As a countermeasure, renovation of the dams and lowering the dam body to enhance the sluicing of sediment were planned [29]. Retrofitting of Saigo Dam, the second from the lower side of Mimikawa River was finished in 2017, and then the operation was started. In the revised operation,

the water level during floods was decreased to enhance the scrubbing and transportation of the settled sediment on the upstream side of the dam body. Due to those operations, sand bars found especially on the downward section of Saigo Dam in recent years have diverse grain sizes even within each sand bar. Besides, this sluicing operation is one of the few examples of this type of operation in Japan and abroad, and it is expected that the riverbed condition will change significantly in a short period. Therefore, along with the development of this method, it is also intended to contribute to the assessment of the impact of the sluicing operation through continuous monitoring while adapting it. Furthermore, in this study, we focused only on particle size as the first step to evaluate the image classification method according to fractions categories by using CNN. In other words, the objective to be classified is only the area of land above the water surface on the day of photographing, and it does not include the area under the water surface where the color change is affected by water absorption and adhesion to the material surface.

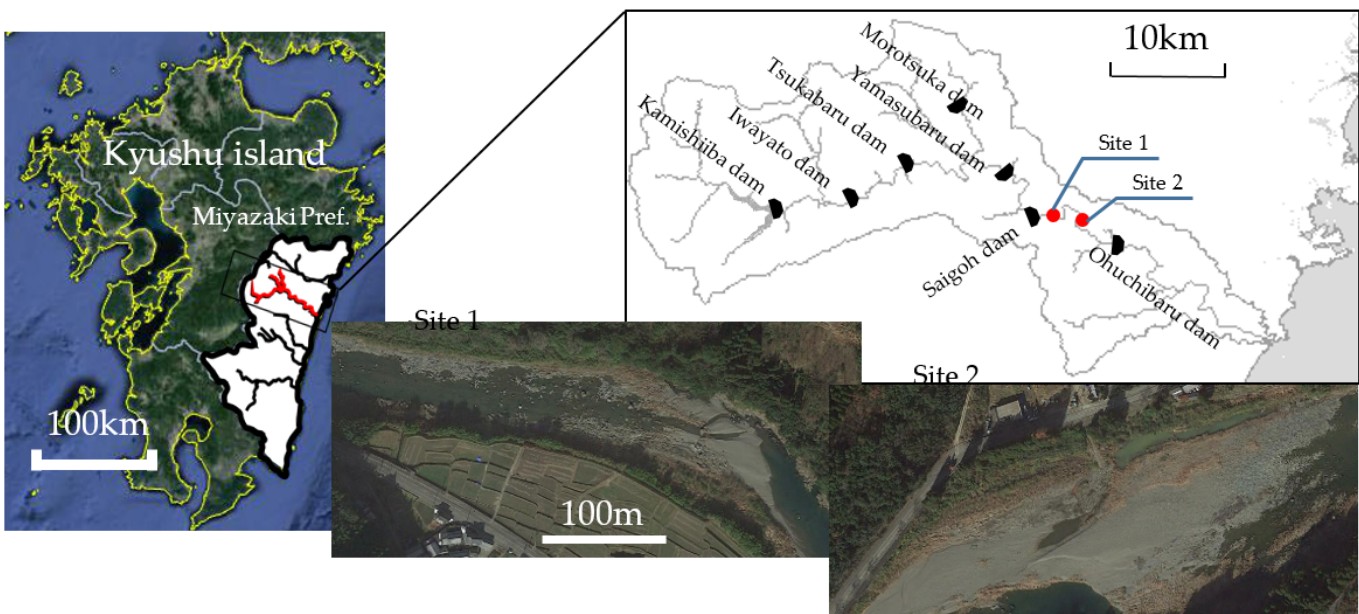

**Figure 1.** Location of the study site on the Mimikawa River in Miyazaki, Japan.

### 3.2. Aerial Photography

The UAV used for aerial photography was a Phantom 4 Pro, manufactured by DJI. Aerial photography from UAV was conducted at 91 points in total over the sand bars. Pictures were taken from 10 m flight altitudes. The camera mounted on the UAV has no optical zoom function, thus that the flight altitude decides the resolution of the images. A lower flight altitude is desirable to improve the resolution, but since the shooting range at one time is limited, the number of shootings is extremely large and unrealistic when trying to observe a wide range. These flight altitudes were decided for such practical reasons. We selected the measurement points where the spatial distribution of constituent materials within the target zone did not vary widely. Figure 2 shows the images taken from UAV in height 10 m from the ground surface. The pictures were taken at 4 different times and dates to verify the effect of sunlight angle on analysis results. The first flight was conducted at 9:00 (1st period, Solar altitude: 24 degrees) and the second at 13:00 (2nd period, 37 degrees) on 13 November 2019. Additionally, in 2020, the field measurement and aerial photography were carried out at 13:00 on 4 September (3rd period, 62 degrees) and at 10:00 on 18 October (4th period, 39 degrees). Both dates in 2020 were cloudy while the date in 2019 was sunny.

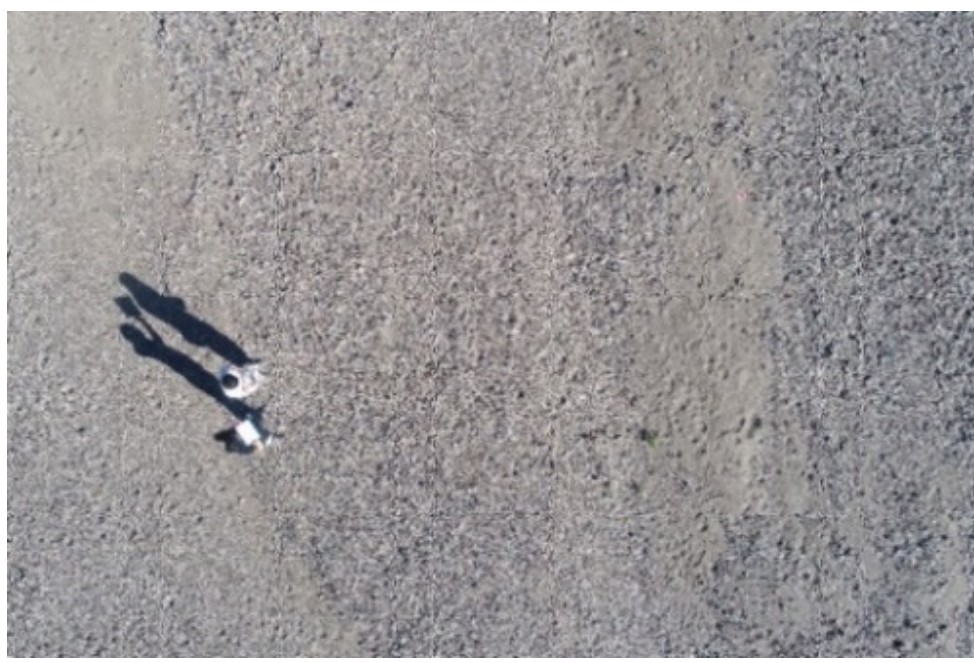

**Figure 2.** Images taken from UAV (10 m).

### 3.3. Sieving

At the same 70 points, the particle size distribution of bed material was measured by sieving and weight scale. Mesh sizes of the sieves were 75, 53, 37.5, 26.5, 19, 9.5, 4.75, and 2 mm (JIS Z 8801-1976) that have intervals considering the logarithmic axis of the graph showing the cumulative curve of particle size. Riverbed material in a 0.5 m × 0.5 m quadrat with a depth of around 10 cm was sampled. Sieving and weighing were carried out in the field.

### 3.4. Image Preprocessing (Clipping)

Although the sampling was performed in a 0.5 m × 0.5 m square by the volumetric method, in this study, the application area of the proposed method was set to a 1.0 m × 1.0 m square centered on the 0.5 m × 0.5 m square. This is because stones with a size of about 20 cm could be seen at some points, and the image range of 50 cm square may not be large enough. Specifications of the camera follow: image size was 5472 pixels × 3648 pixels, lens focal length was 8.8 mm, and sensor size was 13.2 mm × 8.8 mm. When photographing at an altitude of 10 m using this camera, the image resolution was about 2.74 mm/pixel. Therefore, 365 pixels × 365 pixels (equivalent to 1.0 m × 1.0 m square) were trimmed from the original image.

### 3.5. BASEGRAIN Analysis

Only 78 data points of particle size distribution were obtained by the volumetric method. This number was insufficient for training data for the calibration of the parameters in CNN and for test data for evaluating the accuracy of CNN tuned in this study. Then, we prepared the data using BASEGRAIN. We analyzed the surrounding area (8 Yellow areas), which differed from the quadrat (Red area), using BASEGRAIN (Figure 3).

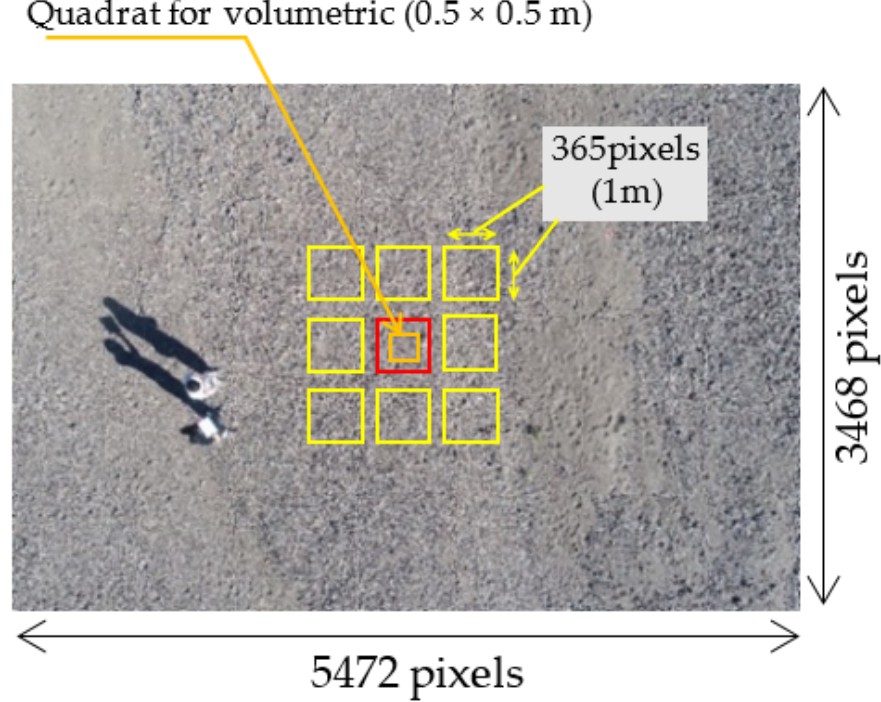

**Figure 3.** Image preprocessing and how to increase the image data. (Red: volumetric method and BASEGRAIN; Yellow: BASEGRAIN only).

*3.6. CNN*

The image recognition code was built with MATLAB®. The CNNs were installed in the code as a module. CNN consist of input and output layers, as well as several hidden layers. The hidden part of CNN is the combination of convolutional layers, pooling layers realizing the extraction of visual signs, and a fully connected classifier, which is a perceptron that processes the features obtained on the previous layers [30]. (Figure 4).

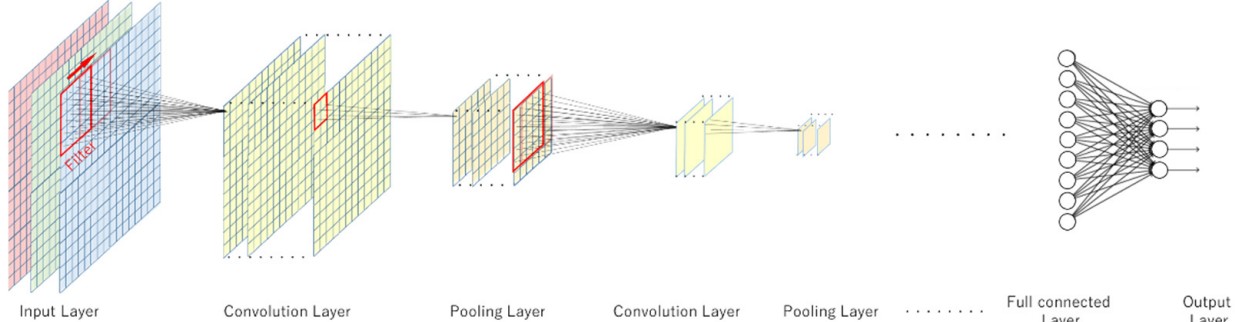

**Figure 4.** Illustration of the convolution neural network.

AlexNet (2012) [30], GoogleNet (2014) [31], VGG (2014) [32], and ResNet (2015) [33] are the major networks that have achieved excellent results in ImageNet Large Scale Visual Recognition Challenge of respective years. The networks trained on ImageNet [34] can classify images into 1000 object categories, such as a keyboard, mouse, pencil, and many animals. They have learned different feature representations for a wide range of images. The details of each network architecture were explained in the references and are not described here.

The networks can be retrained to perform a new task using transfer learning [35]. Transfer learning is commonly used in deep learning applications. The original pre-trained network is used as a starting point to learn the new task. Fine-tuning a network with

transfer learning is usually much faster and easier than training a network from scratch with randomly initialized weights. The features learned before transfer learning can be quickly transferred to the new task using a smaller number of training images.

In this study, we set 3 new categories (Table 1) and tried image classification based on this classification criteria. The 4 networks were evaluated for this study. Further discussion was carried out with the network with the best performance.

## 4. Results

### 4.1. Characteristics of the Surface Condition

Examples of the aerial pictures of the study sites, extracted from the original images with image size the same as that of the quadrat in which the sieving survey was conducted, are shown in Figure 5. In the sand bar, there are diverse conditions on the surface. Even if the median particle sizes were the same, some cases show different uniformity. Figure 6 shows the relationship between the median particle size (D50) and uniformity coefficient calculated from the cumulative curves of the particle distribution histogram for the data of the sieving survey at 78 points. The uniformity coefficient that indicates the variety in particle sizes in mixed natural soils is as follows.

$$U_c = D_{60}/D_{10} \tag{1}$$

where $D_{60}$ and $D_{10}$ are the sieve size through which 60% and 10% (by weight) of the material passes, respectively. It is unity for a material whose particles are all of the same size, and it increases with variety in size. Site 1 is just downstream of Saigo Dam thus the particle size is uniform but composed of coarse gravel; Site 2 shows fine sand accumulation and non-uniformity with larger stones in some points.

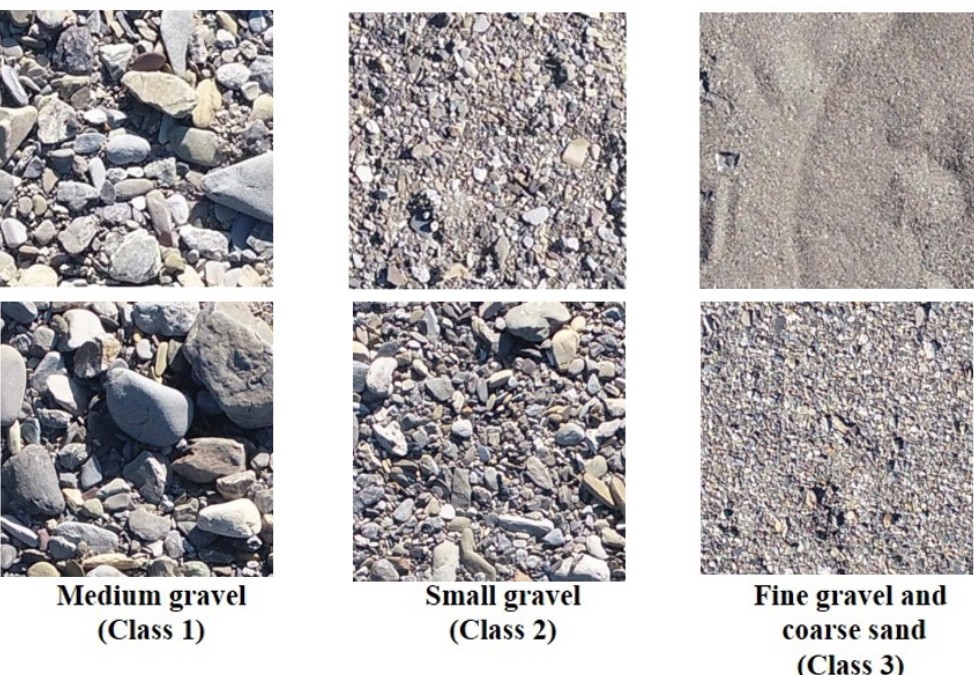

**Figure 5.** Aerial pictures in the quadrats from UAV.

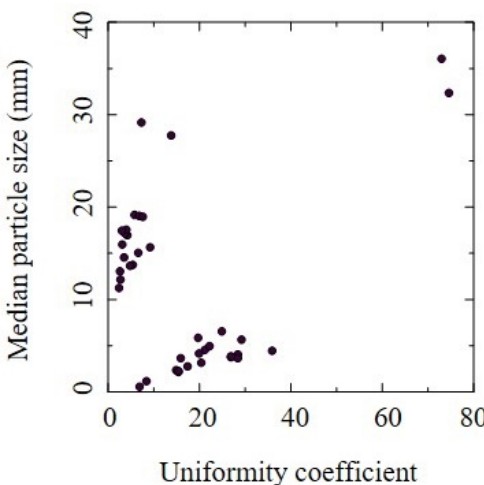

**Figure 6.** Relationship between median particle size and uniformity coefficient.

### 4.2. Application of BASEGRAIN to Study Site

A comparison of the results of the BASEGRAIN analysis and the actual measurement by the volumetric method is shown in Figure 7. The fine particles (average size 2 mm or less) are not plotted because analysis by BASEGRAIN was not possible. The coarse particles (average size 25 mm or more) tend to be underestimated with BASEGRAIN. In the image analysis by BASEGRAIN, each stone was discriminated from an image, and its minor and major axes were measured. When an image with a larger error was checked, it was found that a stone to be analyzed was shadowed by a neighboring stone. Thereby, the part excluding the shadow was recognized as one stone. As a result, it is considered that underestimation occurs in images containing coarse stones. The medium particles (average size 2–24.5 mm) show overestimation with BASEGRAIN. The software has a limit on the size of recognizable particles. The unrecognizable particle size range lower than the resolution (<2.74 mm) is extrapolated from the recognized particle size distribution according to Fehr's methodology approached via a Fuller curve estimation [36]. The methodology needs to assume the ratio of the unrecognized particles. Here, we fixed the ratio of 25%, the default setting of the software. Therefore, an error is likely to occur when the number of fine particles increases.

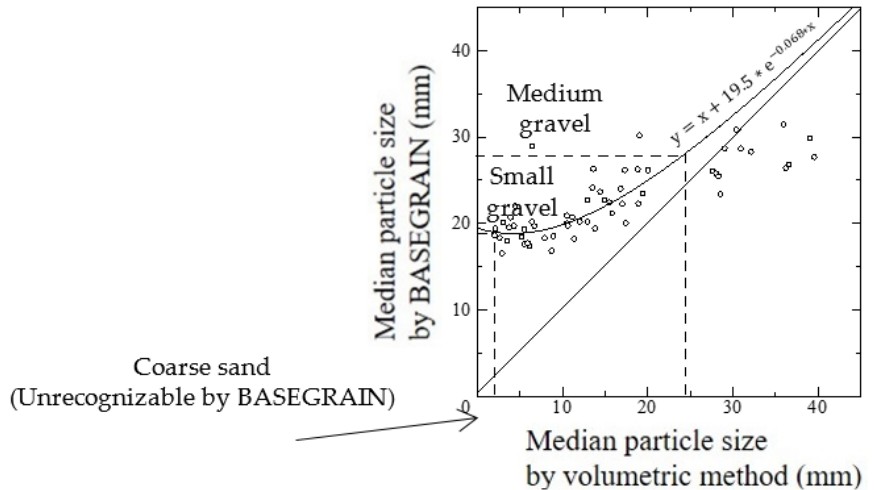

**Figure 7.** Comparison between results for the BASEGRAIN and volumetric methods.

In consideration of the above characteristics, the results of the particle size analysis with BASEGRAIN were corrected by the following procedure and classified into three

categories. When the number of particles recognized by BASE GRAIN was small, it was classified as Class 3 (<2 mm: Fine Gravel and coarse sand). If there was a sufficient number of recognizable particles, the interpolation curve as shown in Figure 7 was applied. Since the thresholds of Class 1 and Class 2 were 24.5 mm in the evaluation by the volumetric method, the corresponding results by BASE GRAIN of 28.2 mm were the threshold for the classification.

An example of a comparison of the particle size accumulation curve between the volumetric measurement in the quadrat area, BASEGRAIN in the quadrat area, and BASEGRAIN in the eight surrounding areas is shown in Figure 8. Although there was an error with the measured value, the results of BASEGRAIN were very similar. Considering that flow velocity at discharge is a factor in determining the distribution of riverbed material, it is reasonable that the flow regime and the particle size of riverbed material remain relatively uniform within a narrow area that fits within the same image. However, images with different particle size characteristics, extremely large stones or obstacles, and inconsistent curvature were not used, even within the same image with a yellow area in Figure 3. In the case that a Yellow area in Figure 3 was classified by BASEGRAIN to the different class from that of Red area, we did not use the image of the Yellow area as test data because it was expected that both might be the same class with the above reason.

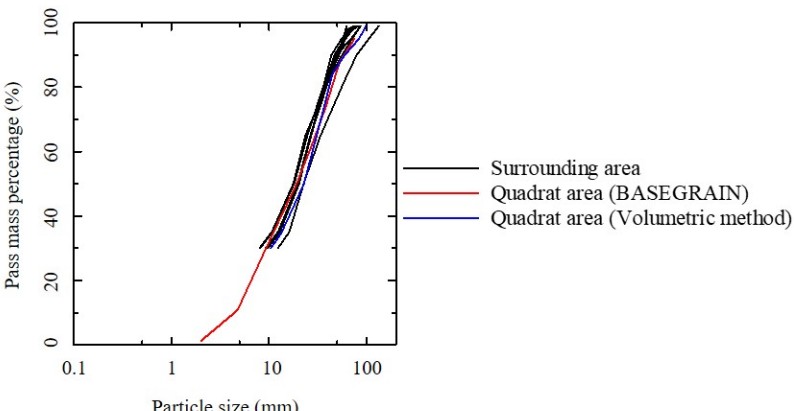

**Figure 8.** Comparison of the particle size accumulation curve by BASEGRAIN and volumetric methods.

### 4.3. Application of CNN

For retraining, 70 images were used, and 435 images were prepared for test data for classification. In the previous studies applying transfer learning [22], there were cases where more images were given as training data. For the annotation set of this study, we prepared images of 70 points measured by the volumetric method and 435 images classified based on the results of the analysis with BASEGRAIN. From the images classified based on the analysis of BASEGRAIN, the image that included objects other than bed material such as timber, grass, etc., were excluded from the training data. The neural networks were trained with a batch size of 10. The several epochs from 12 to 27 were examined to find the optimal number for each network and check overfittings. The training and test were carried out with the single CPU of Intel Xeon E5-1650 v3 3.5 GHz and 64 GB RAM. Training time with the 78 images was only a few minutes.

The accuracies of the classification results were assessed by a multiclass confusion matrix [37]. Overall accuracy varied from 75.4% to 95.4%, depending on the different networks and epochs. As the best result, the case by Google Net with 21 epochs was shown in Table 2. Tables 3–5 are the best results of AlexNet, VGG16, and ResNet, respectively. Trials with each CNNs did not show significant reductions of the accuracies with the increase of epoch number after the optimum numbers thus that overfittings were not observed. The row of the matrix in Tables 2–5 represents the instances in an actual class by the volumetric method, while each column represents the instances in a predicted class by CNN. Recall (producer's accuracy) is how often the references of each class are

classified correctly, while Precision (user's accuracy) is the rate of the correct result among the predicted class. Overall accuracy is the rate of correctly classified images against the number of all images. The overall accuracy of the result with GoogleNet was 95.4%, and the recall in each class of 95.1%, 94.3%, and 98.3% for Classes 1–3, respectively. Even though the resolution of the images was about 2.74 mm/pixel, Class 2 and 3 images whose threshold was 2 mm, which is close to the resolution, were successfully classified.

**Table 2.** The classification by GoogleNet with 21 epoch.

|  | **Class 1** | **Class 2** | **Class 3** | **Recall** | **F-Score** |
|---|---|---|---|---|---|
| Class 1 | 116 | 5 | 1 | 95.1% | 94.7% |
| Class 2 | 7 | 115 | 0 | 94.3% | 94.7% |
| Class 3 | 0 | 1 | 60 | 98.3% | 98.3% |
| Precision | 94.3% | 95.0% | 98.4% |  |  |
| Micro Prec. |  |  |  | 95.4% |  |
| Macro Prec. |  |  |  | 95.9% |  |
| Micro recall |  |  |  | 95.4% |  |
| Macro recall |  |  |  | 95.9% |  |
| Overall Acc. |  |  |  | 95.4% |  |
| Average Acc. |  |  |  | 96.4% |  |

**Table 3.** The classification by AlexNet with 21 epoch.

|  | **Class 1** | **Class 2** | **Class 3** | **Recall** | **F-Score** |
|---|---|---|---|---|---|
| Class 1 | 100 | 22 | 0 | 82.0% | 88.9% |
| Class 2 | 3 | 119 | 0 | 84.4% | 90.4% |
| Class 3 | 0 | 0 | 61 | 100.0% | 100.0% |
| Precision | 97.1% | 84.4% | 100.0% |  |  |
| Micro Prec. |  |  |  | 91.8% |  |
| Macro Prec. |  |  |  | 93.9% |  |
| Micro recall |  |  |  | 91.8% |  |
| Macro recall |  |  |  | 93.1% |  |
| Overall Acc. |  |  |  | 91.8% |  |
| Average Acc. |  |  |  | 93.8% |  |

**Table 4.** The classification byVGG16 with 21 epoch.

|  | **Class 1** | **Class 2** | **Class 3** | **Recall** | **F-Score** |
|---|---|---|---|---|---|
| Class 1 | 110 | 12 | 0 | 90.2% | 87.6% |
| Class 2 | 19 | 103 | 0 | 84.4% | 84.4% |
| Class 3 | 0 | 7 | 54 | 88.5% | 93.9% |
| Precision | 85.3% | 84.4% | 100.0% |  |  |
| Micro Prec. |  |  |  | 87.5% |  |
| Macro Prec. |  |  |  | 89.9% |  |
| Micro recall |  |  |  | 87.5% |  |
| Macro recall |  |  |  | 87.7% |  |
| Overall Acc. |  |  |  | 87.5% |  |
| Average Acc. |  |  |  | 89.9% |  |

**Table 5.** The classification by ResNet with 18 epoch.

|  | **Class 1** | **Class 2** | **Class 3** | **Recall** | **F-Score** |
|---|---|---|---|---|---|
| Class 1 | 92 | 30 | 0 | 75.4% | 85.2% |
| Class 2 | 2 | 119 | 1 | 97.5% | 85.6% |
| Class 3 | 0 | 7 | 54 | 88.5% | 93.1% |
| Precision | 97.9% | 76.3% | 98.2% |  |  |
| Micro Prec. |  |  |  | 86.8% |  |
| Macro Prec. |  |  |  | 90.8% |  |
| Micro recall |  |  |  | 86.9% |  |
| Macro recall |  |  |  | 87.2% |  |
| Overall Acc. |  |  |  | 86.9% |  |
| Average Acc. |  |  |  | 91.4% |  |

GoogleNet could classify the test data into these 2 classes with higher accuracy. Class 3 could be clearly distinguished by any of the 4 networks. In the following discussion, GoogleNet trained with 21 epoch was used. Even the other 3 networks also showed enough accuracy thus that our proposed system, classifying the aerial images from UAV by CNN can give reliable results. However, the accuracies with the other 3 networks were lower than GoogleNet. Especially the misclassifications that occurred between Class 1 and 2. In the following discussion, the reason for this error was examined.

## 5. Discussion

The recognition errors happened between Class 1 and Class2 in all of the results with the 4 types of networks. In the analysis with BASEGRAIN, as mentioned in the Results section, the results were affected by the shadows on stones. Figure 9 is the comparison between the "with shadows" images (taken in the morning: the 1st period) and "no or fewer shadows" images (taken in the afternoon: the 2nd period and under the cloudy condition: the 3rd and 4th period). The extracted ranges of the images are not the same, but the positions of UAV are close to each other, and the conditions on the ground surface, such as microtopography, are similar. Even with our naked eyes, the differences of the shadows were noticed.

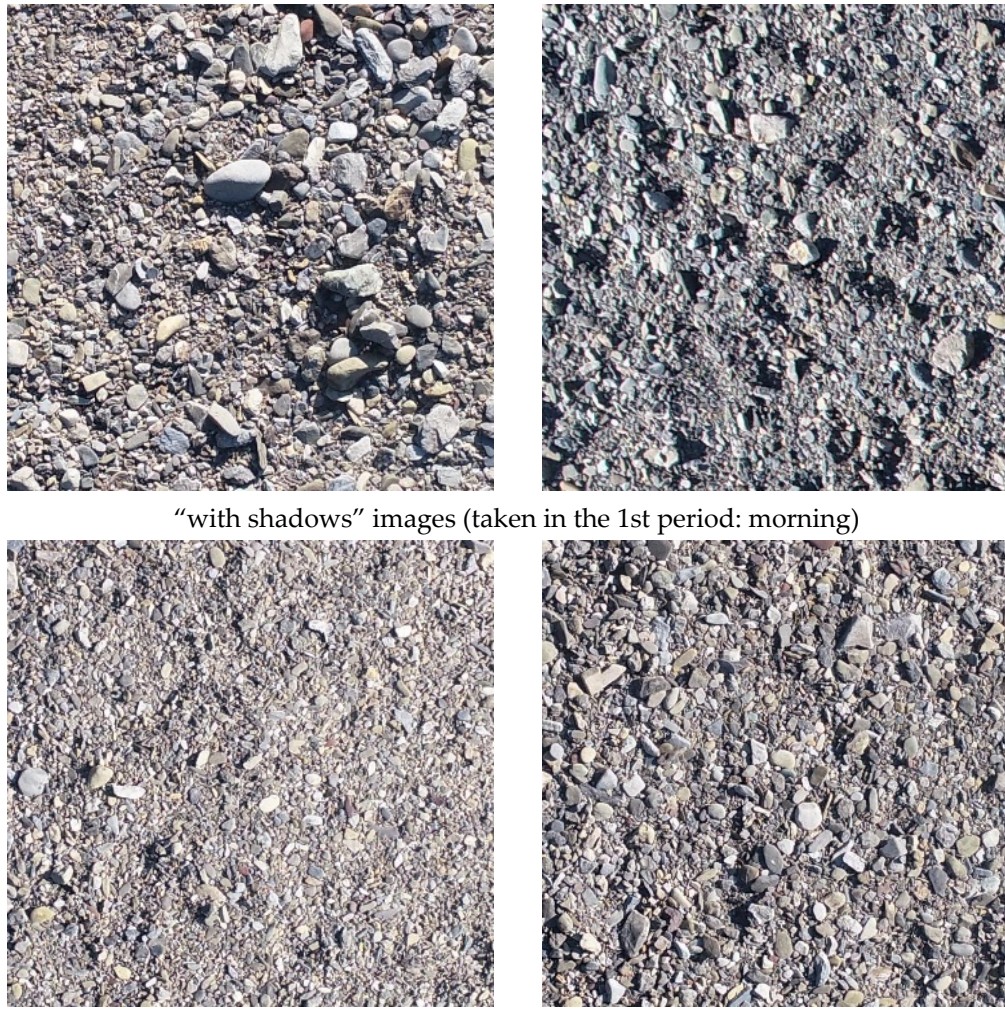

"with shadows" images (taken in the 1st period: morning)

"no or fewer shadows" images (taken in the 2nd period: afternoon)

**Figure 9.** Comparison of the images taken in the 1st and 2nd periods.

As an experiment by changing from the above annotation data and result, the CNN was trained only with the "with shadows" images, then assessed only with the test data of

the "with shadows" images. The confusion matrix for this trial was shown in Table 6. The 55 images evaluated as Class 1 by the volumetric method were miss-classified to Class 2 by the CNN. In the misclassified images, small shadows that might be recognized as fragments were distributed behind stones over the image.

**Table 6.** Confusion matrix assessing CNN trained with "with shadows" images, tested with "with shadows".

|  | Class 1 | Class 2 | Class 3 | Recall | F-Score |
|---|---|---|---|---|---|
| Class 1 | 6 | 55 | 0 | 9.8% | 17.9% |
| Class 2 | 0 | 61 | 0 | 100.0% | 65.6% |
| Class 3 | 0 | 9 | 52 | 85.2% | 92.0% |
| Precision | 100.0% | 48.8% | 100.0% |  |  |
| Micro Prec. |  |  |  | 65.0% |  |
| Macro Prec. |  |  |  | 82.9% |  |
| Micro recall |  |  |  | 65.0% |  |
| Macro recall |  |  |  | 65.0% |  |
| Overall Acc. |  |  |  | 65.0% |  |
| Average Acc. |  |  |  | 82.9% |  |

As for comparisons, all the combinations of "with shadows" and "no or fewer shadows" images for training and test data were examined, shown in Tables 7–9. The network trained with "with shadows" images has lower accuracy while that trained with "no or fewer shadows" could correctly recognize the images even that has the different features of shadows. By learning from the pattern without shadows, the network could detect the parts that have a similar pattern from the "with shadows" images.

**Table 7.** Confusion matrix assessing CNN trained with "with shadows" images, tested with "no or fewer shadows".

|  | Class 1 | Class 2 | Class 3 | Recall | F-Score |
|---|---|---|---|---|---|
| Class 1 | 5 | 53 | 3 | 8.2% | 15.1% |
| Class 2 | 0 | 61 | 0 | 100.0% | 68.9% |
| Class 3 | 0 | 2 | 59 | 96.7 | 95.9% |
| Precision | 100.0% | 52.6% | 95.2% |  |  |
| Micro Prec. |  |  |  | 68.3% |  |
| Macro Prec. |  |  |  | 82.6% |  |
| Micro recall |  |  |  | 68.3% |  |
| Macro recall |  |  |  | 68.3% |  |
| Overall Acc. |  |  |  | 68.3% |  |
| Average Acc. |  |  |  | 84.2% |  |

**Table 8.** Confusion matrix assessing CNN trained with "no or fewer shadows" images, tested with "with shadows".

|  | Class 1 | Class 2 | Class 3 | Recall | F-Score |
|---|---|---|---|---|---|
| Class 1 | 60 | 1 | 0 | 98.4% | 93.8% |
| Class 2 | 7 | 54 | 0 | 88.5% | 90.8% |
| Class 3 | 0 | 3 | 58 | 95.1 | 97.5% |
| Precision | 89.6% | 93.1% | 100.0% |  |  |
| Micro Prec. |  |  |  | 93.4% |  |
| Macro Prec. |  |  |  | 94.2% |  |
| Micro recall |  |  |  | 94.0% |  |
| Macro recall |  |  |  | 94.0% |  |
| Overall Acc. |  |  |  | 94.0% |  |
| Average Acc. |  |  |  | 94.0% |  |

**Table 9.** Confusion matrix assessing CNN trained with "no or fewer shadows" images, tested with "no or fewer shadows".

|  | Class 1 | Class 2 | Class 3 | Recall | F-Score |
|---|---|---|---|---|---|
| Class 1 | 61 | 0 | 0 | 100.0% | 100.0% |
| Class 2 | 0 | 61 | 0 | 100.0% | 100.0% |
| Class 3 | 0 | 1 | 60 | 98.4 | 99.1% |
| Precision | 100.0% | 98.4% | 100.0% | | |
| | Micro Prec. | | | 99.4% | |
| | Macro Prec. | | | 99.4% | |
| | Micro recall | | | 99.4% | |
| | Macro recall | | | 99.4% | |
| | Overall Acc. | | | 99.4% | |
| | Average Acc. | | | 99.4% | |

On the other hand, the random shadows on the training data might disturb the pattern and reduced the accuracy of the network. Therefore, it is important to use images that do not contain shadows for the training to improve the accuracy of the network. Recently, Shadow Detection and Shadow Removal techniques were developed [38,39]. In further studies, the application of these processes before the classification of riverbed images will be considered.

## 6. Conclusions

The CNN image recognition was used to observe particle sizes of riverbed material. The CNN successfully classified the images based on image features with high accuracy into the three categories: fine gravel and coarse sand, small gravel, and medium gravel without discriminating particles and measuring the diameter. GoogleNet showed the best performance among the 4 networks we examined. Shadows in images affected the classification results. To reduce the error caused by shadows, the images without shadows must be used for training data. Here in this study, the survey was carried out at 2 sites in a stream. The Color and morphology of stones are comparatively uniform. Thus that, to improve the classification accuracy, the amount of training data of the variety of images from different streams is needed. Besides, as a future challenge, we will consider the underwater application of the proposed method.

In combination with the location information embedded in the images taken from UAV, the spatial distribution of the bed material particle size in a wide river bank area can be observed. This precise information could not be provided by the conventional methods requiring much labor, time, and cost. The proposed method will contribute to environmental protection and flood control in river channels.

**Author Contributions:** Conceptualization, M.I.; methodology, H.T. and M.I.; software, H.T. and M.I.; validation, H.T. and S.A.; formal analysis, H.T., S.A. and M.I.; investigation, M.I.; resources, M.I.; data curation, H.T., S.A. and M.I.; writing—original draft preparation, H.T.; writing—review and editing, M.I.; visualization, H.T.; supervision, M.I.; project administration, M.I.; funding acquisition, M.I. All authors have read and agreed to the published version of the manuscript.

**Funding:** This study was funded by JSPS KAKENHI Grant Number 17H03314.

**Conflicts of Interest:** The authors declare no conflict of interest.

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
