# Peer review of "Differentiation of River Sediments Fractions in UAV Aerial Images by Convolution Neural Network"

_remotesensing, doi:10.3390/rs13163188_

Round 1

Reviewer 1 Report

Dear Authors, 

Thank you for addressing all my comments and I don't have any further comments. The paper is accepted from my side. 

Best Regards

Author Response

Thanks for your helpful comments and suggestions throughout the whole review process.

Reviewer 2 Report

Dear authors,

thank you for explanation of comments in my review and I can say that the article was improved a lot and its current form is very interesting for readers. Method is easy to apply wherever above water. Considering effect of shadows, I cannot imagine how this method could work under water for such small objects. However, many studies exist (e.g. https://ieeexplore.ieee.org/document/8600472; https://ieeexplore.ieee.org/abstract/document/8242527; https://www.sciencedirect.com/science/article/abs/pii/S0262885619301362 ....) and it will be possible to adopt approach also for underwater conditions.

  • line 72: "This study has similar to the spatial scale and resolution of our case." better would be: ".. is similar to the spatial scale ..."
  • line 197: I recommend reword "the scale on the image" because in line 322 a term "image resolution" is used in the same context - decide which term you will use in the article
  • line 270: I suggest to reword this sentence on error of extrapolated data and explain in detail accordingly to the achieved results. Fine gravel and corse sand belong the 3rd category which was also tested,

kind regards, Reviewer

Author Response

Thanks for your helpful comments and suggestions throughout the whole reviewing process. I followed your suggestions in this term and revised my manuscript. The articles you introduced will be helpful for my further studies. 

Line 72

Thanks for your pointing out. I changed to “This study is…”

Line 197

Thanks for your suggestion. I reworded to “the image resolution” for uniform expression.

Line 270

I referred to the methodology of the extrapolation and explained the assumption of a ratio of unrecognizable fine particles is required. Thanks again for your suggestion.

This manuscript is a resubmission of an earlier submission. The following is a list of the peer review reports and author responses from that submission.

Round 1

Reviewer 1 Report

Dear Authors,

I have reviewed your paper and please find the attached file for my comments. 

Best Regards 

Author Response

Dear the reviewer

Thanks for your helpful comment. We tried to revise our manuscript following to your comments. I prepared my answers to your comments as below.

  1. Please add result details in the abstract section.

Thanks for your suggestion. I add the concrete description of the result details in the abstract

  1. The length of the introduction section is wide and it is better to separate the introduction
    section into related work sections.

Thanks for your suggestion. I separated the introduction section to the 2 sections

  1. Please add a full stop at the end of the sentences (line 48)

I add the punctuation at the end

  1. The structure of the introduction section is not in a proper format. It is a mixed form of
    related work and introduction. The authors should update the introduction part.

Thanks for your suggestion. I reform the introduction section

  1. The related work from this paper is not strong to claim the necessity of this research. The
    authors should add a detailed analysis of existing approaches and specify the importance of
    this research and how this research overcomes the existing challenges.

Thanks for your comment. Our study use the property of CNN extracting feature of images but without measuring diameters of particles extracted from an image. Only classifying riverbed images to the categories defined by median particle size. Line 73-85 in our revised manuscript described it with some additional references.

  1. Please add a paper contribution to the end of the introduction section.

I put the paper contribution before the references. I followed the format provided by MDPI

  1. Please add “The rest of the paper is organized as ….’ details at the end of the introduction
    section.

The organization has already be evident from the sectioning of this short communication. I like to omit the suggested description.

  1. In subsection 2.2, it is better to show some image samples from UAV for better
    understanding.

I added some images from UAV at different Flight altitudes.

  1. It is better to show the pictorial representation of subsection 2.4 (Image pre-processing).

Image pre-processing is just clipping. I add the pixel numbers in Figure 3.

  1. The authors should explain the CNN model details clearly. Most of the important details are
    missing from the paper. Please consider the hyperparameter values and out of the CNN
    model etc.

Thanks for your suggestion. We add the values of Batch, Epoch and training time with the CPU.

  1. The authors used transfer learning for their model training. However, it is difficult to
    understand the importance of transfer learning in this content. Please add important
    references related to the CNN model, transfer learning, etc. The authors should improve
    subsection 2.6 CNN.

Thanks for your suggestion. We add the important references and explanation of CNN and Transfer learning. In the section of CNN and Introduction.

  1. Please add a connection point in the Figure 4 CNN model. Please refer to some CNN models
    and update Figure 4.

I updated  Figure 4.

  1. Please add a small description at the starting of Section 3.

I think we don’t need any description at the start of result.

  1. The number of images used for training and testing is not sufficient for the validation of the
    system. The authors should consider the large amount of data set for their system’s
    validation.

For increasing the validation data, we need to use BASEGRAIN for the surrounding area of the quadrat zone observed by the volumetric method. Far from the quadrat zone, similarity to the condition of the center getting lower, and confidence of the analysis with BASEGRAIN getting lower. That is why we only use the extracted images just beside the quadrat zone for BASEGRAIN analysis. On the other hand, as mentioned in the introduction, the volume metric method needs hard work. In the near future we will increase the number of volumetric survey but different river. As mentioned in the conclusion, we are interested in the applicability to the rivers that has a different characteristics of the riverbed material. We like to spend our labor to that also. We understood that the number of the image is short so that we submit this article as communication.

  1. The authors should claim why the CNN model gives better results than other neural
    networks. Please add a different model analysis.

Thanks for your suggestion. We suppose the important point of this manuscript is finding the applicability of CNN with transfer learning to riverbed observation though with only limited number of training data.

  1. The authors should consider CNN hyperparameter tuning, training, and testing time and
    should claim how the CNN model gives accurate results than existing approaches.

Due to the small number of training images, hyperparameter can have only narrow range. That didn’t make a big difference.

  1. The experiment and result section should improve in terms of results. The current results
    from the paper are not sufficient for paper acceptance. The authors should validate their
    system with relevant results.

As a communication, we aim to reported a limited results but rapidly.

  1. Please add relevant references related to the CNN model.

We add the reference of GoogLeNet and the others.

  1. If it is possible, the authors can also think about the confusion matrix analysis, accuracy, loss,
    precision, recall, and F1- score parameters for CNN model analysis. (To make the experiment
    and results section strong).

We add indexes of confusion matrix

  1. Finally, the authors should improve the English language and sentence contraction. The
    paper has major flows in English writing

Thanks for you recommendation. We passed a proof reading.

Reviewer 2 Report

Line92:” which is used in this study, is a machine learning method that is attracting 92 attention because of its superior performance in the image recognition field.”

Superior is already a conclusion. Please elaborate superior to what

or try to use more balanced phrase showing zero prejudice in advance.

The method has shown in some key studies to have results higher than ( 80-%  90%  ????Kappa  value >0.85  maybe ????what are measurements to come to a conclusion “”superior”)

“affinity for the classification based on the multispectral data like Landsat and Senti-98 nel[15,16].

Maybe mention something on scale differences here. The study   [17] is UAV  thus similar to your scale and appropriate.

250For these images, the percentage of correct an-250 swers is high. However, images in the afternoon have small shadows even if including 251 large stones, and result in a low percentage of correct answers

Shadows are notorious in photogrammetric analysis. In this case, a dedicated time slot ( noon plus minus 2 hours) or similar could be recommended with also more precise seasonal details. Maybe autumn winter images are not too be used at all. Morning is very broad try to define using solar angle and season .

I recommend to slpit results away from discussion

Plaeas concentrate chapter  only results

Make chapter 4 discussion

And move conclusions to a chapter 5

There is a lot of industrial image analysis. Mostly in the environment of https://www.mvtec.com/  machine vision.

They study particles from industrial machines, chips and detritus , concrete and cement analysis etc. etc.

Please add some review not from remote sensing alone but also from the world of industrial machine vision.

Make very sure you split important discussion topics and create chapter 4 ONLY discussion, keep chapter 3 for results only. Conclusions moved to chapter 5

Author Response

Dear the reviewer

Thanks for your helpful comment. We tried to revise our manuscript following to your comments. I prepared my answers to your comments as below.

Line92:” which is used in this study, is a machine learning method that is attracting 92 attention because of its superior performance in the image recognition field.”

 Superior is already a conclusion. Please elaborate superior to what or try to use more balanced phrase showing zero prejudice in advance.

 The method has shown in some key studies to have results higher than ( 80-%  90%  ????Kappa  value >0.85  maybe ????what are measurements to come to a conclusion “”superior”)

 We used the word of “Superior” to CNN and the references, not for our results. Generally, CNN is recognized as Superior for Image recognition. We erase that word from our manuscript to avoid such misunderstanding. Thanks for your comment.

“affinity for the classification based on the multispectral data like Landsat and Senti-98 nel[15,16].

Maybe mention something on scale differences here. The study   [17] is UAV  thus similar to your scale and appropriate.

However, those are the application of CNN in remote sensing field. We would like to keep those as the references while I described that those are different in resolution and scale.

250For these images, the percentage of correct an-250 swers is high. However, images in the afternoon have small shadows even if including 251 large stones, and result in a low percentage of correct answers

Shadows are notorious in photogrammetric analysis. In this case, a dedicated time slot ( noon plus minus 2 hours) or similar could be recommended with also more precise seasonal details. Maybe autumn winter images are not too be used at all. Morning is very broad try to define using solar angle and season . I recommend to slpit results away from discussion Plaeas concentrate chapter only results

Make chapter 4 discussion

Thanks for your comments.  The date we carried out our survey was 15 Nov. So that Solar altitude in that season was quite low but the result was acceptable (not superior..).  We add the information of solar altitude at the flight time.

In addition, we considered that the shadow also can be a feature. We add our opinion line 315-319 of revised manuscript

There is a lot of industrial image analysis. Mostly in the environment of https://www.mvtec.com/  machine vision.

They study particles from industrial machines, chips and detritus , concrete and cement analysis etc. etc.

Please add some review not from remote sensing alone but also from the world of industrial machine vision.

Thanks for your suggestion, we add the references from industrial section and medicinal section. 

Make very sure you split important discussion topics and create chapter 4 ONLY discussion, keep chapter 3 for results only. Conclusions moved to chapter 5

Thanks for your suggestion. We arranged the sectioning

Reviewer 3 Report

The paper entitled “Application of image identification by Convolution Neural Network to aerial images from UAV for riverbed broad monitoring" proposed the use of CNNs for the classification of three types of riverbed rocks, where the images used for training the CNN were taken at two different time zones and covering different places of a riverbed. 

They used a drone at different altitude for acquiring the images and Matlab for training the CNN. 

My remarks are: 

The state-of-the-art about image processing and machine learning applied to riverbed monitoring needs to be improved, it will be better to provide a more recent advance in this emerging field.

The weakest point of the work is the application of a CNN, trained with fixed architecture, and the little detail about the training time, epochs, batch size and all the data related to a common machine learning literature is missing. It will be very helpful to provide more analysis of the differences between the images of different time zones since they provided a trial between training the CNN with images of one time zone and tested on different time zone, concluding that is  better to be used only images with the same time-zone, but it might be a bias towards this conclusion for not pre-processing well the data.  

Related to the data provided in section 2.4, it is not clear how they conclude about the Ground Sampling Distance and if they applied some image calibration before taking the images.

Additional remarks are the following.

line 22   "we constructed a CNN based on training data... "
I do not understand the sentence, whether you constructed a CNN as a Neural Architecture Search based on training data or you have projected a CNN as a classifier?

line 34   I suggest adding some info about the interpretation of riverbed materials related to the flow of the river. 

line 80   It is 22x43 pixels?

line 91   typo?

line 147  It is not clear if some Zoom was applied or why did they choose to take images at 5 or 10 m. Additional info stating at 162 is better to be added in the aerial photography section.

line 190  Rephrase, then the model can be restructured is not well accurate, the model maintains the structure but not the weights, and transfer learning is a well-known methodology for overcoming the little amount of data in computer vision, it will be useful to check the concept of fine-tuning as well. You can talk a little bit more about the architecture of GoogleNet.

line 204  It is not clear what the uniformity coefficient is telling related to figure 6. 

line 156  Why these values since you have used only 3 classes..?

line 246  Reference for producer-s accuracy?

line 242  Why you have used only 90 images for training and 280 images for test?  What about the usual partition of data in Machine learning?  Also, what is the number of validation data? For how long was trained the model? Also it is confusing on how did you get this 373 images since you cover 39 spatial points.   

line 254  To improve accuracy you have used only a set of images of the morning, right? There is a mismatch in the distribution of the images. It is necessary to provide some figure examples of the same place but with different times. Did you apply some standardization or normalization of the images before feeding them to the NN to combat the mismatch distribution of luminosity? Was the training dataset for the second trial balanced?  Did you also train the model with images from the second time zone and test with both datasets? 

          Matlab also has some functions to produce a proper confusion matrix, it might be helpful also to provide other metrics as Recall and F1, usually presented in classification problems.

Author Response

Dear the reviewer

Thanks for your helpful comment. We tried to revise our manuscript following to your comments. I prepared my answers to your comments as below.

line 22   "we constructed a CNN based on training data... "
I do not understand the sentence, whether you constructed a CNN as a Neural Architecture Search based on training data or you have projected a CNN as a classifier?

Thanks for your comment. The description was inappropriate. We correct the words.

line 34   I suggest adding some info about the interpretation of riverbed materials related to the flow of the river. 

Thanks for your comments. We add some literatures interpretation of riverbed materials related to the flow of the river[3,4]. 

line 80   It is 22x43 pixels?

To avoid such misunderstanding, we clearly describe it (Line 122 of the revised draft)

line 91   typo?

Sorry for our short expression. We add the explanation.

 (Line 79-81 of the revised draft)

line 147  It is not clear if some Zoom was applied or why did they choose to take images at 5 or 10 m. Additional info stating at 162 is better to be added in the aerial photography section.

We didn’t explain the reason. We add it Line168-172

line 190  Rephrase, then the model can be restructured is not well accurate, the model maintains the structure but not the weights, and transfer learning is a well-known methodology for overcoming the little amount of data in computer vision, it will be useful to check the concept of fine-tuning as well. You can talk a little bit more about the architecture of GoogleNet.

Thanks for your comment. Our description was not correct. We add some explanation about GoogLeNet

line 204  It is not clear what the uniformity coefficient is telling related to figure 6. 

We add the explanation of Uniformity Coefficient Line 240

line 156  Why these values since you have used only 3 classes..?

Due to the condition of our study site. This short communication is our first try. We will rapidly move to next challenge it in near future.

line 246  Reference for producer-s accuracy?

Thanks for your comment. We add the reference of confusion matrix totally.

line 242  Why you have used only 90 images for training and 280 images for test?  What about the usual partition of data in Machine learning?  Also, what is the number of validation data? For how long was trained the model? Also it is confusing on how did you get this 373 images since you cover 39 spatial points.   

Larger number of images is better for training. Usually number of validation data is less than training data.

In our case, we have 2 kinds of images, evaluated by volumetric method and BASEGRAIN.

Due to the hard work, we could not obtain so many images with volume metric method. We add the images whose particle size distribution was evaluated by BASEGRAIN. For training data, the quality of the images is also important matter. We supposed the images for training should have clear feature, surely categorized to one class. So that we use the images with volume metric method (red area in Figure. 3) and the images around it(Yellow in Figure3). If the BASEGARAIN result of the Yellow zone images is not same class as the red zone, we stopped the use of it for training. On the other hand, test data contains the images the feature judged with our naked eyes was not clear.

line 254  To improve accuracy you have used only a set of images of the morning, right? There is a mismatch in the distribution of the images. It is necessary to provide some figure examples of the same place but with different times. Did you apply some standardization or normalization of the images before feeding them to the NN to combat the mismatch distribution of luminosity? Was the training dataset for the second trial balanced?  Did you also train the model with images from the second time zone and test with both datasets? 

CNN did not measure the particle size, just grasp as “feature”. In my understanding, the shades found in the images of A.M. is the typical feature of the images for class 1.

We also tried standardization and normalization but could not significant improvement in the results.

          Matlab also has some functions to produce a proper confusion matrix, it might be helpful also to provide other metrics as Recall and F1, usually presented in classification problems.

We add the othe parameters of the confusion matrix.

Reviewer 4 Report

The present article approaches the issue of monitoring riverbed material which has a high importance in the river ecosystems.

Reading only the abstract makes very difficult to identify which are the main results of this article. The abstract should be rephrased in order to emphasize the main results and the main conclusion.

The full article doesn't clarify the shortcomings mentioned in the abstract.

The selection of BASEGRAIN as analysis software in this article doesn't seems to be the best option since the riverbed materials are much complex, from different perspectives, than grains.

There are also several other questions that need answers:

Which are the main results of this article? Which is the reliability level of the methodology proposed by you?

You should focus more on the methodology section, results and discussions section as well as on the conclusions.

Major revision is required.

Author Response

Dear the reviewer

Thanks for your helpful comment. We tried to revise our manuscript following to your comments. I prepared my answers to your comments as below.

Reading only the abstract makes very difficult to identify which are the main results of this article. The abstract should be rephrased in order to emphasize the main results and the main conclusion.

The full article doesn't clarify the shortcomings mentioned in the abstract.

Thanks for your comment. We revised the abstract

The selection of BASEGRAIN as analysis software in this article doesn't seems to be the best option since the riverbed materials are much complex, from different perspectives, than grains.

For increasing the training and test data, we need to use BASEGRAIN for the surrounding area of the quadrat zone observed by the volume metric method. We use only the images of the surrounding area where the results of BASEGRAIN is similar to the BASEGRAIN result of the quadrat zone (distribution was observed by volume metric). With that process, the nominated surrounding areas are same class as the quadrat zone on the center. The other images different from the quadrat one were not used for training and test. We carefully used the images analyzed by BASEGRAIN.

Which are the main results of this article? Which is the reliability level of the methodology proposed by you?

The riverbed condition can be evaluated by CNN, this short communication reported rapidly.

You should focus more on the methodology section, results and discussions section as well as on the conclusions.

Perhaps the important point of our communication was not expressed clearly. Our method does not measure the particle size directly, just finds features of the images by CNN. We don’t know clearly what is the feature for each class (that is the typical characteristics of CNN). Especially by the transfer learning, with only limited number of the images, we could train the CNN to the high accuracy. This method omit the direct measurement of the particles so that it can classify the image that dominant grains are tiny beyond the recognition as independent particle. We revised the sentences in the introduction and methods explaining clearly that point.

Reviewer 5 Report

Dear authors,

The Communication solves from my point of view, very interesting topic of the river sediments classification using remote sensing technologies - a remotely piloted aircraft system, in this article called UAV. The usage of UAV for the river sediments mapping and classification is very practical, as the authors proved in the introduction and proved with recent studies where conservative “hard-work” field survey techniques were applied to measure fractions of river sediments. Each technology which reduces time and labour spent in the field saves money which can be used more efficiently. Besides, these practical positive issues of the applied research I can say that the Communication does not bring any novelty to the basic research to explain “How CNN works in case of sediments fractions” in its current version. Generally, the methodical part is weak. In the methodology, I expected that specifics of a filter/s used to create a feature map and the process of CNN training would be explained in detail not only by general characteristic (also citation is missing in the methodical part). Further for instance, explanation is missing how shades equalizing was performed during algorithms of CNN or an explanation of the difference between producers and discrimination accuracy is missing. How were both kinds of accuracies calculated? On the other hand, the introduction explains details which are not necessary for this kind of the work. Communication is a short but comprehensive message about something new that the authors would like to announce to the public. This Communication announced the application of the methods which is not clearly explained and properly documented in results.

Please find detailed comments in attached PDF file: remotesensing-1144916-peer-review-v1

and bellow arranged in lines respectively:

  • Title: reword the title - it sounds strange – similarly like application of identified image - for me it makes no sense. Try simple like Recognition/Discrimination/Identification/Differentiation of river sediments fractions in UAV aerial images by Convolution Neural Network. Monitoring was not presented in results. Therefore, shall not be in the title.
  • 20-21: Reword the sentence, divide to 2 sentences, it is hard to read.
  • 25: Change a keyword "channel bed condition" to e.g. "image analysis" "discrimination", in case of the usage of the word "discrimination" please exactly use it only in certain cases.
  • 98: Consider this reference because satellite images provide totally different resolution not suitable for gravel sediments survey.
  • Table 1: Use the same as in text "fine gravel and coarse sand".
  • 117: Not necessary, this workflow is simple and does not require additional image.
  • In my opinion it is “sediment fractions discrimination method in UAV imagery by CNN” not “an image discrimination method”. In case you insist on "the image discrimination method" please cite and describe in several words/2-3 sentences how it works.
  • The terms discrimination, discriminated ... shall be used in the whole article only in relation to the river sediments classification according to fractions categories - unify the terminology in the Communication.
  • Figure 2: use a simple physical geographic map or another map with general information, satellite maps are not suitable because we can see only green and shadow zones and cannot distinguish land cover objects in this general small scale.
  • 165: But fraction of coarse sand was limited to 0.4 mm, explain in results.
  • 180-181: cite references
  • 182-183: Explain briefly basis of CNN in the introduction or methodology including citation/s. For instance, terms like: feature map, producers´ accuracy, discrimination accuracy - what is the difference between both accuracies?
  • 183: reword
  • 186: The statement "The CNN improves the discrimination accuracy ... " is not clear; The statement must be supported with cited references and it is necessary to say: the discrimination accuracy of what? image? river sediments? fractions of river sediments? Please, unify terminology.
  • 189: cite from GoogleNet or whatever relevant. Was any plugin installed to web browser when GoogleNet was applied? If yes, note a reference link. It would be interesting for a reader which object categories were used from the object library - like it was presented at the ImageNet treemap of images.
  • 193: A problem might be the last finest category, Considering the camera resolution. Therefore, the authors must say how it affected results and it is OK.
  • Figure 4: It would be more efficient to adopt this scheme to the questioned river sediments fractions and specify CNN steps in this context because the current scheme does not bring any new information about CNN.
  • Figure 5: It is from Dron?
  • 203-205, Figure 6, 217, 219, 226, Figure 7 and 8: calculation formula is missing in the methodology
  • 232-233: What does it mean? Where a yellow frame is explained in methods?
  • 243-244: Explain how discrimination and producer´s accuracies were measured in the methodology.
  • 248: It is not documented in the method and results - document or remove.
  • 249: It is not documented in the results- document or remove. Which details of Class 1 can I check? Where?
  • 252: Firstly, prove effect of shadows and then conclude this statement.
  • Table 2: How these values in matrix were calculated? Add minimally to the methods minimally citation referring to accuracy assessment in CNN.
  • Table 3: Did you mean morning and afternoon?
  • 266: This Communication does not bring any specific new aspects to the application of CNN. The main problem is weak methodology. Shading affect is not properly proved with in results - how shades equalization worked in CNN? Specify whatever what would be helpful to differ between more and less shaded fractions of river sediments. There exist some experiments: DOI: 10.1109/TIP.2019.2904267 etc.
  • References: Adjust using MDPI reference style.

regards,

Reviewer

Author Response

Dear the reviewer

Thanks for your very helpful comments. We are researcher in river engineering and beginner for AI and image identification. We like to enhance to use CNN and other AI tools for our research. Following to your comments we revised our manuscript.

  • Title: reword the title - it sounds strange – similarly like application of identified image - for me it makes no sense. Try simple like Recognition/Discrimination/Identification/Differentiation of river sediments fractions in UAV aerial images by Convolution Neural Network. Monitoring was not presented in results. Therefore, shall not be in the title.

Thanks for your suggestion. We like to change the title as you suggested.

  • 20-21: Reword the sentence, divide to 2 sentences, it is hard to read.

We revised the abstract to describe the main result concretely with shortening the sentence.

  • 25: Change a keyword "channel bed condition" to e.g. "image analysis" "discrimination", in case of the usage of the word "discrimination" please exactly use it only in certain cases.

In my opinion, this communication aims to observe channel bed condition with Image recognition and UAV. That’s the target of remote sensing study. I like to keep this term as a keyword. The main topic or purpose of this communication is not the development of CNN, as you pointed out.

  • 98: Consider this reference because satellite images provide totally different resolution not suitable for gravel sediments survey.

I like to list these references as the example of CNN application in remote sensing field. I understood input data of these references is not RGB image but values of multispectral sensors in a pixel. That is different from image recognition finding features in a image. I listed the application of CNN and image recognition in the other fields. As a contrast, I like to keep it.

  • Table 1: Use the same as in text "fine gravel and coarse sand".

Thanks for your comment. I revised the term.

  • 117: Not necessary, this workflow is simple and does not require additional image. In my opinion it is “sediment fractions discrimination method in UAV imagery by CNN” not “an image discrimination method”. In case you insist on "the image discrimination method" please cite and describe in several words/2-3 sentences how it works.

I omitted the Figure. That was no need.

  • The terms discrimination, discriminated ... shall be used in the whole article only in relation to the river sediments classification according to fractions categories - unify the terminology in the Communication.

I use “classification” basically for our method in the revised manuscript. Just for the case of references, identifying an object in an image, we use “discriminate” or discrimination.

  • Figure 2: use a simple physical geographic map or another map with general information, satellite maps are not suitable because we can see only green and shadow zones and cannot distinguish land cover objects in this general small scale.

Thanks for your suggestion. We change the map to simple.

  • 165: But fraction of coarse sand was limited to 4 mm, explain in results.

We add the explanation in the result LINE309 in our revised manuscript. Thanks for your suggestion emphasizing the result.

  • 180-181: cite references
  • 182-183: Explain briefly basis of CNN in the introduction or methodology including citation/s. For instance, terms like: feature map, producers´ accuracy, discrimination accuracy - what is the difference between both accuracies?

We cite the reference of CNN [31]

  • 183: reword
  • 186: The statement "The CNN improves the discrimination accuracy ... " is not clear; The statement must be supported with cited references and it is necessary to say: the discrimination accuracy of what? image? river sediments? fractions of river sediments? Please, unify terminology.

We referred the expression in the reference.

  • 189: cite from GoogleNet or whatever relevant. Was any plugin installed to web browser when GoogleNet was applied? If yes, note a reference link. It would be interesting for a reader which object categories were used from the object library - like it was presented at the ImageNet treemap of images.

That’s not the plugin of web browser. I don’t know such version.

GoogLeNet was trained with Imagenet, I add it on the revised manuscript.

  • 193: A problem might be the last finest category, Considering the camera resolution. Therefore, the authors must say how it affected results and it is OK.

We don’t insist to distinguish the fine particles. Just classify images based on the features or patterns detected by CNN. In my understanding, comparative size of finest particle against resolution is not so important.

  • Figure 4: It would be more efficient to adopt this scheme to the questioned river sediments fractions and specify CNN steps in this context because the current scheme does not bring any new information about CNN.

The purpose of this communication is not bringing any new information about CNN. Just an application to the new field.

  • Figure 5: It is from Dron?

Yes it is.

  • 203-205, Figure 6, 217, 219, 226, Figure 7 and 8: calculation formula is missing in the methodology

My explanation was too short, I added the explanation to uniformity coefficient.

  • 232-233: What does it mean? Where a yellow frame is explained in methods?

I missed to point the figure No. I added it to the revised manuscript.

  • 243-244: Explain how discrimination and producer´s accuracies were measured in the methodology.

I cited the reference[34].

  • 248: It is not documented in the method and results - document or remove.
  • 249: It is not documented in the results- document or remove. Which details of Class 1 can I check? Where?

The explanation was not clear. We tried to document it again in the new manuscript

  • 252: Firstly, prove effect of shadows and then conclude this statement.

We add the statements

  • Table 2: How these values in matrix were calculated? Add minimally to the methods minimally citation referring to accuracy assessment in CNN.

Thanks for your comment, we cite.

  • Table 3: Did you mean morning and afternoon?

The expression was not clear, I revised it.

  • 266: This Communication does not bring any specific new aspects to the application of CNN. The main problem is weak methodology. Shading affect is not properly proved with in results - how shades equalization worked in CNN? Specify whatever what would be helpful to differ between more and less shaded fractions of river sediments. There exist some experiments: DOI: 10.1109/TIP.2019.2904267 etc.

Thanks for your introducing the interesting reference. In my understanding, the introduced reference is the method for detecting shadow.

We revised the explanation in the section of discussion, newly sectioned following to another reviewer. We cannot catch up what is the feature detected by CNN, but by changing the combination of training data and test data, we can guess some effect of shades.

  • References: Adjust using MDPI reference style.

Thanks again for your very informative review comments.

We did our best for this revision even while we cannot satisfy you.

Round 2

Reviewer 1 Report

Dear Authors,

Thank you for addressing my concerns. However, the authors didn't address all my concerns (Comments: 6, 7, 13, 15, 16, and 17). The authors didn’t understand my concern 6, the reviewer asked about the paper contributions, not the author's contributions. Also, the novelty of the paper is not sufficient to accept the paper for publication. In the case of reviewer’s concern 15, the authors should try different deep learning models like VGG, ResNet, Inception, etc. to validate their proposed system (Some image classification models). I believe, the authors didn’t understand my previous comments. In the paper, the experiment and result section is not strong for acceptance of the paper for a high impact factor journal (IF: 4.5), even the paper is a communication type. I strongly recommend to the authors to find a low impact factor journal (less than 2, based on their current results) for their next submission.

Best Regards

Reviewer

Author Response

Dear the reviewer

Thanks for your comments on my revised manuscript in the 1st round and sorry for misunderstanding your comments.

To comment 6: Please add a paper contribution to the end of the introduction section.

I apologize firstly for my misunderstanding with your comment. The contribution to the remote sensing field or river management has already described in the upper part of my abstract. Usually, I don’t describe it at the end of the abstract.

To comment 15: The authors should claim why the CNN model gives better results than other neural networks. Please add a different model analysis.

Thanks for your suggestion. We will carry out the trial with other CNN in further study. The aim of this communication is the applicability of image recognition technology to river bed monitoring. We can find a lot of published studies of the application trial of a model to a specific purpose (ex. [22] in our manuscript). For focusing on the application, using one model is simple and clear. In fact, the other reviewers accepted our approach though they gave the related comments in the 1st round.

One of the reasons that we started with GoogLeNET is the low risk of overfitting. The number of our training data was limited, so that overtraining should be considered. Perhaps VGG and ResNET also satisfy that requirement. We will try it in the next stage.

Reviewer 3 Report

Dear authors,

For this round, I am glad to see the improvements in the manuscript. Certainly, some issues had been explained, some references were added but still, I do not see the convenience of ref 15-18 to name the application of CNNs in medicine and so on. Maybe some review about the field would be enough.

I am not sure if the notation in line 302 is adequate. Some typos in line 324 and it might be in other lines as well. Legend in the Figure 3, it is difficult to be seen. 

The Discussion can be improved, since lines 309 to 312 are not accurate, understanding the interpretability of CNNs is a recent field of study and it is not just by analyzing the nature of the data. Now, they were presented more details about the training of the CNN that inlight some other questions about why it was only trained for 6 epochs, why only 30 images per epoch, how about overfitting, etc.

Even though, I understand that you have applied a CNN to this particular classification task and the focus of the paper is to show the feasibility of the application. Regardless of the field of work of the authors, I would like to encourage a multidisciplinary work, to see the application of other better architectures of CNNs (like VGG, ResNet, and others) to show even better results than this exploratory work. I would like to recommend the publication of this present article despite some issues that might be still worked out.

Author Response

Dear the reviewer

Thanks for your helpful comments. I tried to improve the manuscript based on your comments.

Certainly, some issues had been explained, some references were added but still, I do not see the convenience of ref 15-18 to name the application of CNNs in medicine and so on. Maybe some review about the field would be enough.

Thanks for your comments. I reflected on the comments from the other reviewer in the 1st round on that part. I like to keep it in this round with the respect to the comments.

I am not sure if the notation in line 302 is adequate.

We improved the expression. See Line 301 in the revised manuscript

Some typos in line 324 and it might be in other lines as well.

Thank you for your accurate suggestion. The structure was not logical. We tried to improve it.

 Legend in Figure 3, it is difficult to be seen. 

Perhaps yellow characters with the background of stones were not clear. The legend with solid gray background is placed with a slight sift.

The Discussion can be improved, since lines 309 to 312 are not accurate, understanding the interpretability of CNNs is a recent field of study and it is not just by analyzing the nature of the data.

Thanks for your helpful suggestion. We added the cites [35]&[36]. Perhaps the riverbed images are kind of texture. That might be the reason why we could get good identification results. We could find the study discussing the better affinity of CNN to texture recognition than shape identification[36].

 Now, they were presented with more details about the training of the CNN that inlight some other questions about why it was only trained for 6 epochs, why only 30 images per epoch, how about overfitting, etc.

Thanks for your understanding of the restrictions in our field observations. We will continue to increase the volume of data for more accurate differentiation.

Even though, I understand that you have applied a CNN to this particular classification task and the focus of the paper is to show the feasibility of the application. Regardless of the field of work of the authors, I would like to encourage a multidisciplinary work, to see the application of other better architectures of CNNs (like VGG, ResNet, and others) to show even better results than this exploratory work. I would like to recommend the publication of this present article despite some issues that might be still worked out.

Thanks for your comments to encourage us. I understood that we need more discussion about the other applications soon later. We wish to proceed to the next step.

Reviewer 4 Report

No other comments to make.

Author Response

Dear the reviewer

Thanks for your contribution to our submission. We revised our manuscript reflecting the comments from the other reviewers. We are very glad for your review and comments in this round again.

Regards

Reviewer 5 Report

Dear authors,

The Communication was considerably improved, and uncertain issues explained or supporting references were inserted. The quality of the manuscript raised. I accept the usage of a key word “Channel bed condition”, you are right because this keyword navigates readers to a scientific field where CNN was applied. I appreciate, that terminology and application of image classification using multispectral data from different resources – ranging from satellites to Drons was clearly explained from a broader perspective. Nevertheless, a reference to “a novel AI disease-staging system for grading diabetic retinopathy” is far beyond the investigated field of the sediments fractions. At this stage, it is your choice to decide if this reference is relevant. Graphic in updated Figure 1 is much better; now, the study site is recognizable, and it explains well the object of interest. Figure 4 was also markedly improved. Methodical part is clear and good ground for results. Discussion reflects the main aim “employing the CNN image recognition to observe particle sizes of riverbed material”. I recommend the Communication for the publication.

Please find minor comments in attached PDF file: remotesensing-1144916-peer-review-v2

and bellow arranged in lines respectively:

  • 24-25: I can understand that a context of “time zone” is different from the world time zoning (Uniform Time Zone is 8 hours behind Coordinated Universal Time) however the using of “the universal time zone” and “the same time zone” is not clear because the survey was performed during different day phases – as was mentioned morning and afternoon but not in different time zones. The term “time zone” is usually related with the setting of the world time.
  • 283: Which studies did you mean? Cite or explain.
  • Figure 6: If (a), (b) and (c) are not explained in the Figure title then remove from the Figure 5 because it is confusing, a reader tends to search where is explained (a) ...
  • Figure 9: Did you compare the same rectangles?

regards,

Reviewer

Author Response

Dear the reviewer

Thanks for your many thought-provoking comments through round1&2. I could improve the quality of the manuscript based on those.

  • 24-25: I can understand that a context of “time zone” is different from the world time zoning (Uniform Time Zone is 8 hours behind Coordinated Universal Time) however the using of “the universal time zone” and “the same time zone” is not clear because the survey was performed during different day phases – as was mentioned morning and afternoon but not in different time zones. The term “time zone” is usually related with the setting of the world time.

Thanks for your advice. I changed to ‘(temporal) period’ .

“Time zone” was inappropriate for this case indeed.

  • 283: Which studies did you mean? Cite or explain.

I cited [22] that you pointed out, the study in the medical field. I feel the similarity with that study in that approach based only on the feature extraction of the image by CNN and transfer learning without measuring the identified object in the image. In that study, a lot of training data was used.

  • Figure 6: If (a), (b) and (c) are not explained in the Figure title then remove from the Figure 5 because it is confusing, a reader tends to search where is explained (a) ...

Thanks for your suggestion. I omitted extra notations.

  • Figure 9: Did you compare the same rectangles?

Unfortunately not in the same place, but they are close to each other. Reproducing the flight position as the previous flight is quite difficult. The conditions such as microtopography are similar. I add the explanation in Line 320-323
